# Plasma-activated water: Mechanism and treatment duration for postharvest disease control and shelf-life enhancement of mango under ambient storage

Mst. Habiba Tamanna[1], Showdia Sarmin[2], Umme Afroza Irin[1], Mst. Khadiza Khatun[1], Mamunur Rashid Talukder[3], Md. Mahmodol Hasan[1]*

1 Plant Pathology Laboratory, Department of Agronomy and Agricultural Extension, Rajshahi University, Rajshahi, Bangladesh, 2 Plant Pathology Department, Pest Management Division, Bangladesh Jute Research Institute, Dhaka, Bangladesh, 3 Plasma Engineering Laboratory, Department of Electrical and Electronic Engineering, Rajshahi University, Rajshahi, Bangladesh

* mmhasan@ru.ac.bd

## Abstract

In Bangladesh, a large quantity of mango is lost every year due to post-harvest diseases, particularly anthracnose and stem-end rot. Therefore, sustainable post-harvest management is crucial for reducing the losses. In this study, we investigated the effects of plasma treatments on mango to mitigate post-harvest losses. The mangoes were submerged in distilled water (DW), and then DW-submerged mangos were treated for 10 minutes employing multi-capillary bubble discharged plasma jet system using air and oxygen gases separately. Plasma treatments significantly influenced disease incidence, severity, and physio-chemical properties of mangoes. On the 10th day, the Khirsapat mango, treated with a 10-minute air-discharge plasma, exhibited a significant (≤0.05) reduction in anthracnose incidence (20%) and severity (2.33%) compared with control (incidence 80% and severity 61.67%). Similarly, the 10-minute air-discharge plasma treatment consistently reduced the incidence (20%) and severity (15%) of stem-end rot. For the Fazlee variety, the incidence (22%) and severity (3.67%) of anthracnose were reduced compared with the control (89% and 56.67%), while stem-end rot was completely inhibited for up to 10 days under the 10-minute air-plasma treatment. In addition, both mango varieties showed increased total soluble solid (18% and 19%), retained good moisture content (77.82% and 85.34%), but reduced physiological weight loss (3.12% and 8.36%), and extended shelf life (6 days). Firmness degradation was lowest in air plasma treatment (4.20% in Khirsapat and 4% in Fazlee) compared to control (5.76% and 5.79%). It is interesting to note that the plasma treatment of both varieties showed higher mineral contents (K, Ca, Mg, and P), while Vit C declined modestly (16.0 and 25.10 mg/100g) compared to control. Therefore, a 10-minute air-discharge plasma treatment effectively reduced disease incidence and severity through enhancing TSS, mineral

**Data availability statement:** All relevant data are within the manuscript and its Supporting information files.

**Funding:** The author(s) received no specific funding for this work.

**Competing interests:** We declare that we have no known competing financial interests or personal relationships that could have appeared to influence the work reported in this paper. We do confirm that this work is original and has not been published elsewhere, nor is it currently under consideration for publication elsewhere. We have no conflicts of interest to disclose.

contents, physiological properties, and overall storage life that highlighting its potentiality as an eco-friendly postharvest technology.

## Introduction

Mango (*Magnifera indica* L.) is the most popular and widely consumed tropical fruits, universally cherished for its excellent flavor, appealing aroma, and high nutritional value [1–3]. In terms of food value, it contains sufficient amounts of total soluble solids (TSS), minerals, vitamin-C, β-carotene, and dietary fiber [4–6]. It is cultivated commercial in over 40 countries, including Bangladesh, which ranks seventh globally in mango production [7]. Approximately 1842653.78 metric tons of mangos are produced from the 158,420.45 hectare of mango garden in Bangladesh [8]. Like other tropical fruits, mangoes continue their physiological and biological activities even after harvest. Consequently, harvested fruits gradually lose quality and become highly susceptible to postharvest infections, which negatively affect their marketability [1,9,10]. Without proper management, mango fruits —both green and ripe—are vulnerable to attack by fungi, bacteria, and other harmful organisms, resulting in a shortened shelf life. Anthracnose, caused by *Colletotrichum* spp. and stem-end rot, induced by a complex of fungal pathogens, are the two most prevalent and economically significant postharvest diseases accountable for substantial losses of mango [11–15]. Post-harvest losses reach 60% or higher due to anthracnose, while losses exceed 80% [16] from stem-end rot. Anthracnose affects both young and mature fruits, with black sunken necrotic lesions rapidly develop on the fruit surface [17–19]. In contrast, stem-end rot is primarily characterized by dark lesion at the base of the pedicel, which develops into a soft, water-soaked rot extending from the stem end [20,21]. Synthetic fungicides are the primary tools for controlling post-harvest diseases; however, their indiscriminate use promotes fungal resistance, environmental contamination, and toxic residues on fruits, thereby threatening food safety and market profitability [22–24]. To overcome the bottleneck of the existing synthetic fungicide-based technologies, alternative eco-friendly and sustainable method for fruit disinfection is urgently expected. Plasma technology is a cutting-edge approach that is currently being used in the agricultural sectors [25–27]. The plasma activated water (PAW) contains reactive oxygen species (ROS) and reactive nitrogen species (RNS), which interact with the pathogenic microbes without damaging plant tissues or host cells [28–30]. This technology has shown promising results in the decontamination of fruits and vegetables without degrading their qualities [31,32]. Several studies have demonstrated that PAW treatment effectively inactivates fungi, bacteria, yeasts, and other microorganisms on fruits and vegetables and enhance post-harvest quality, including fresh cut iceberg lettuce [33], sprouts [34], Chinese bayberries [35], strawberries [36], grapes [37], fresh-cut kiwi [38], pears [39], fresh-cut apples [40], and lemons [41]. Although the previous investigations have reported disease reduction in fruits in other countries, to the best of our knowledge, no comprehensive studies have assessed using PAW as a post-harvest treatment for mango in Bangladesh, where disease outbreak and post-harvest losses are predominantly high. Moreover,

very limited information is available regarding the effects of PAW on the management of anthracnose and stem end rot, and physicochemical quality attributes of mango during storage. Thus, this study addresses this gap by scrutinizing the ability of PAW to control major post-harvest diseases while conserving fruit quality and extending shelf-life. We hypothesize that PAW treatment could significantly reduce disease incidence in mango without causing adverse effects during storage. Therefore, the present study aims to evaluate plasma treatment as a cutting-edge post-harvest technology to improve fruit quality and extend the shelf life of mango.

## Materials and methods

The experiment was carried out at Plant Pathology Laboratory, Department of Agronomy and Agricultural Extension and Plasma Engineering Laboratory of the Department of Electrical and Electronic Engineering, University of Rajshahi to study the postharvest management of anthracnose and stem-end rot of mango employing multi-capillary bubble discharged plasma jet system using air and oxygen gases separately.

### Experimental design and treatments

The experiment comprised of two mango varieties (Khirsapat and Fazlee) and four treatments, namely: $T_0$ (control), $T_1$ (mangoes submerged in DW and DW was treated with multi-capillary air bubble discharge plasma jet system for the duration of 10 minutes), $T_2$ (mangoes submerged in DW and DW was treated with multi-capillary oxygen bubble discharge plasma jet system for the duration of 10 minutes, and $T_3$ (chemical treatment serving as the positive control). This experiment was laid out using completely randomized design (CRD), in which treatments were randomly assigned to homogeneous experimental units. Under laboratory conditions, the experimental units were homogenous, therefore, blocking was unnecessary. For this reason, the CRD was appropriate for this experiment. Each treatment condition had three replications with five mangoes included in each replication. Therefore, a total of 15 mangoes were used for each treatment condition.

### Mango treatment reactor

The mango treatment reactor was simple in design and consisted of a 0–15 kV (50 Hz) power supply, a 1 L glass beaker, a 3-capillary plasma jet, and air and oxygen gas flow controllers. A schematic diagram of the experimental setup for the mango treatment reactor is shown in Fig 1A. The plasma jet was prepared using three capillary tubes (length 75 mm, inner diameter 0.9 mm). The capillary tubes were attached together, and a copper wire (0.25 mm in diameter) was inserted into each tube to serve as a power electrode. At one end, a 5 mm gap was maintained between the capillary tube opening and the inner copper wire to generate the discharge within this gap. The opposite end of the capillary tube bundle was inserted into the glass tube (100 mm long and inner diameter 10 mm) and sealed with epoxy compound. The inner power conductor was routed through an opening in the wall of the glass tube and connected to an insulated wire, while the opposite end of the glass tube was connected to a high-pressure gas flow line. The inner power conductor was connected to the high-voltage power supply. The mango samples to be treated were placed in a 1L beaker containing 500 mL DW. The plasma jet was immerged 30 mm into DW, and the ground conductor was also submerged into DW. Gas was supplied to the jet at a flow rate of 1L.min$^{-1}$. Upon application of high voltage, plasma was generated within the discharge gap of the capillary tubes. The DW along with submerged mango sample was treated for 10 minutes. After treatment, the mangoes, along with treated water, were sealed in an air tight polybag for 30 minutes. The treated mangoes were then dried under laminar air hood at room temperature. A photograph of the reactor during mango treatment is shown in Fig 1B.

### Determination of plasma properties

Power dissipated in the discharge was calculated by integrating the discharge voltage times current, shown in Fig 2A, over a cycle by $W_{diss} = (\frac{1}{T}) \int_0^T v(t)i(t)dt$. The discharge power was ~3.37W. while, Fig 2B shows the species produced in air

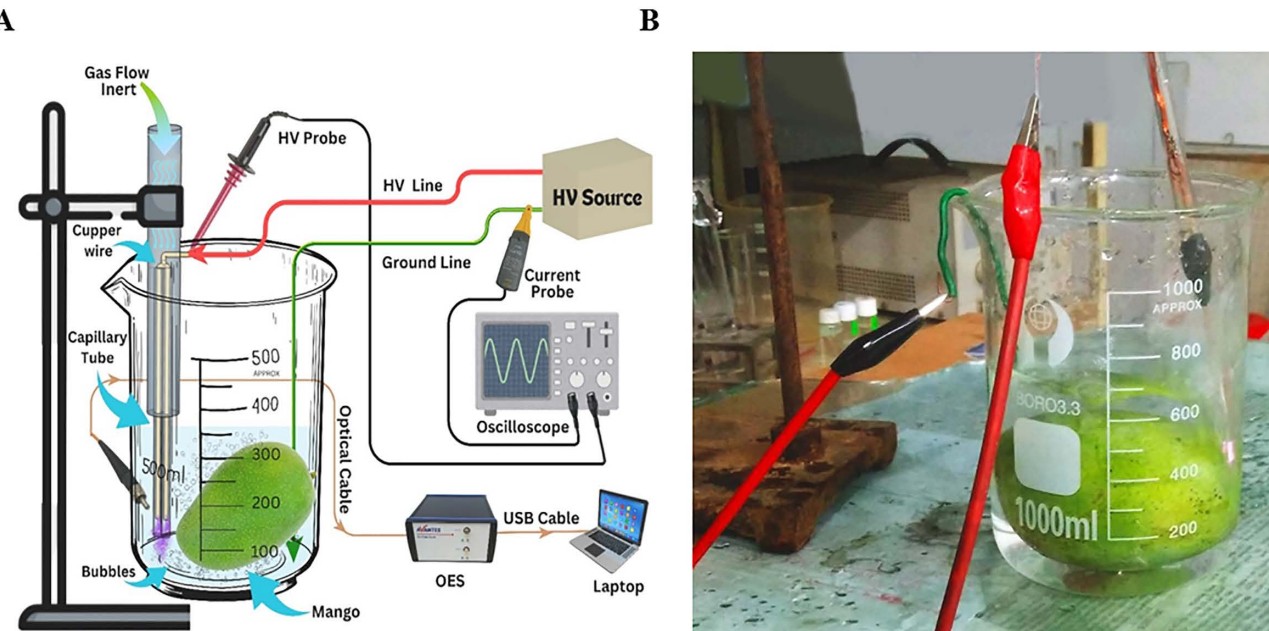

**Fig 1. (A) Schematic of the mango treatment reactor and (B) Photograph of the reactor under mango treatment in DW.**

and oxygen discharges. The dominant plasma species are $N_2$(C-B) and $N_2^+$ (B-X) in air discharge, and OH and O are in the oxygen discharge.

## Fungicidal treatment

The selected mangoes were immerged for 5 minutes in a solution of Amistertop (Dephenoconazole+azoxystrobin) at a concentration of 0.6 ml/L. Amistertop contains a protective strobilurin fungicide (active ingredient of azoxystrobin), which inhibits spore germination of *Colletotrichum gloeosporioides*, and a systemic triazole fungicide (active ingredient of diphenoconazol), which target dormant infections of *Botryodiplodia theobromae*, thereby prolonging shelf-life of both mango varieties [42,43]. This fungicide was selected because it is widely recommended in Bangladesh and approved by the relevant regulatory authorities.

## Isolation and identification of *Colletotrichum gloeosporioides* and *Botryodiplodia theobromae*

The fungi *C. gloeosporioides* and *B. theobromae* were isolated from the infected ripe mangoes showing anthracnose and stem-end rot symptoms (Figs 3 and 4) and were identified based on their color, spore morphology, and mycelial characteristics [44–47].

## Observation of the treatment effects

Visual inspection was carried out daily in order to assess disease incidence, disease severity, maturity stage, taste, color, firmness and shelf life, and data were recorded accordingly. In addition, physicochemical parameters including pH, TSS, moisture content, mineral content, and vitamin-C, were measured using standard analytical methods.

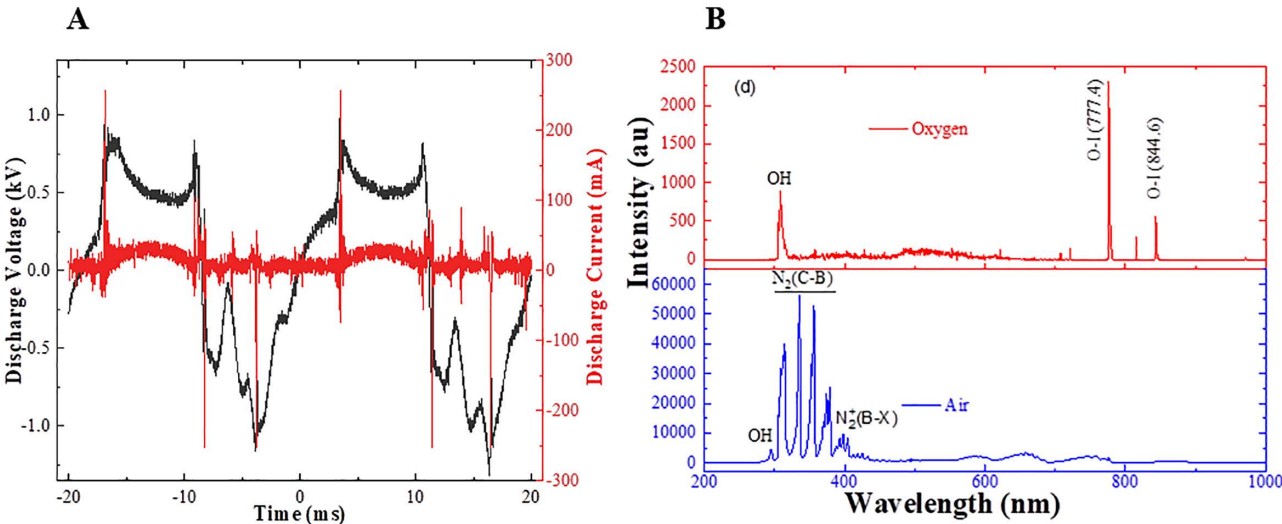

**Fig 2. (A)** Current-voltage characteristics of the plasma discharge used for power dissipated in the discharge. **(B)** Optical emission spectrum of the plasma discharge used to identify species produced in the discharge.

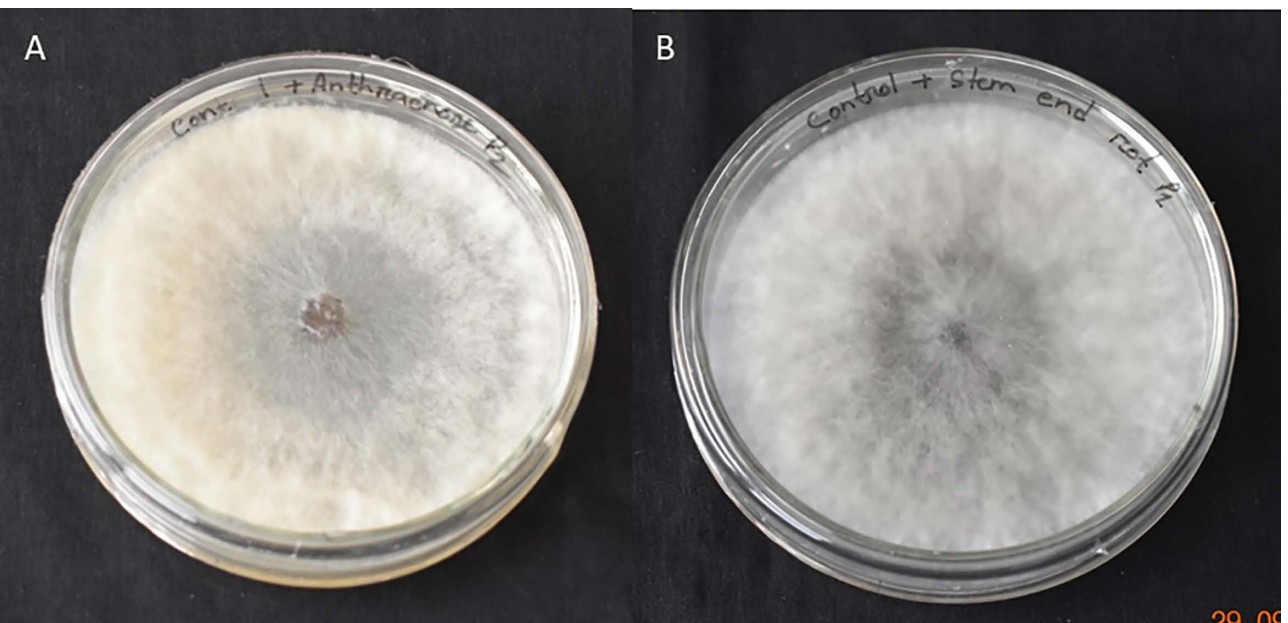

**Fig 3. Photograph showing culture of (A)** *Colletotrichum gloeosporioides* **and (B)** *Botryodiplodia theobromae*.

## Observation of disease incidence

The treated mangoes were placed on sterile brown paper and maintained in the laboratory at $30 \pm 2^0$C. Regular observations were made to monitor any changes in the mangoes for up to 10 days after treatment. Disease incidence was recorded regularly to monitor the development of infection during storage and to accurately determine the onset of visible symptoms. Disease incidence was estimated using the following formula:

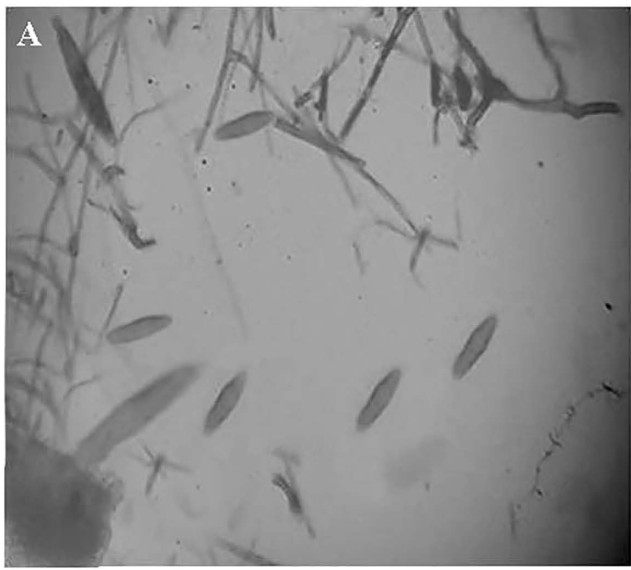 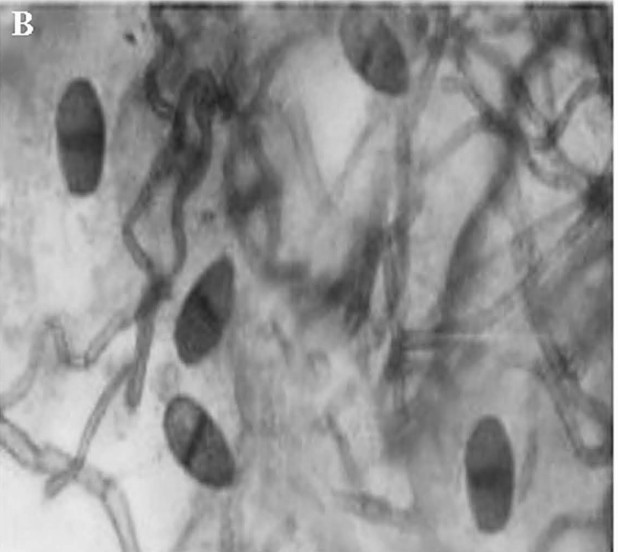

**Fig 4. Photograph showing conidia of (A)** *Colletotrichum gloeosporioides* **and (B)** *Botryodiplodia theobromae***.**

$$\text{Disease incidence (\%)} = \frac{\text{Number of infected mango in each replication}}{\text{Total number of mango in each replication}} \times 100$$

## Observation of disease severity

Disease severity quantifies the proportion of the diseased area relative to the total area of the infected plant part. The percentage of diseased fruit area was calculated using the formula proposed by Johnson [48].

$$\text{Disease severity \% (DS)} = \frac{A_0}{A} \times 100,$$

Where, DS is the percentage of disease severity, $A_0$ is the area of fruit infected by disease, and A is the total area of fruit.

## Determination of pH

$P^H$ of the control and treated mango samples was measured from the juice by a $p^H$ meter (PHS-25 $P^H$ meter) and the corresponding data were recorded.

## Determination of physiological weight loss (%)

Five fruits from each treatment were weighted individually for data collection. The percentage of physiological weight loss (%WS) was calculated using the following formula:

$$\text{Physiological weight loss \ (\%WL)} = \frac{IW - FW}{IW} \times 100,$$

where, WL represents the percentage of total weight (g) loss, and IW and FW are the initial and final weights (g) of fruits, respectively.

## Determination of moisture content (%)

Moisture content (MC) is the quantity of water present in the fruit. The MC (%) of mango fruits was determined following the official AOAC method [49]. The weighing dishes were oven-dried at 105°C for 3 hours before adding 3g of sample. The dishes were then cooled in a desiccator and weighed. Subsequently, the samples were dried at 105°C to a constant weight, cooled again, and reweighed. The moisture content was calculated based on the loss in weight using the following formula:

$$MC(\%) = \frac{W1}{W2} \times 100$$

Where $W_1$ is the initial weight of the sample before drying (g) and $W_2$ is the final weight of the sample after drying (g).

## Determination of total soluble solids (TSS)

The total soluble solids (TSS) for each treatment were determined according to the technique specified by AOAC [50] using a hand-held refractometer (0–80 °Brix). The refractometer was calibrated with double-distilled water at room temperature prior to measurement. Fruit juice was filtered through muslin cloth to remove debris. A drop of the clarified juice was then placed on the prism surface of the refractometer, the cover was gently closed, and the TSS value was record as percentage (°Brix).

## Determination of vitamin C and mineral content

Vitamin C content was determined according to the AOAC method [51] using the AOAC 967.21 2,6-dichlorophenolindophenol (DCIP) titration method. Sample extracts were homogenized in an L-ascorbic acid solution and subsequently titrated against DCIP previously standardized with a USP ascorbic acid reference standard. The titration endpoint was indicated by the appearance of a faint pink color. Vitamin C content was calculated and expressed as mg/100g of sample.

## Estimation of mineral contents

Mineral content was determined using a flame atomic absorption spectrometric method (Thermo-Scientific iCE 3000). For this, treated mango fruits were sliced into small pieces and oven-dried at 70°C, after which the dried samples were ground into a fine powder. The powdered samples were then placed in crucible and ashes in a muffle furnace by gradually increasing the temperature up to 550° C for 6 hours until complete ashing was achieved. Later, 2g of ash was dissolved in 100 ml HCl (0.005M), centrifuged at 5000 rpm for 3 min, and filtered. The resulting solution was analyzed using a flame atomic absorption spectrophotometer (AAS) equipped with an autosampler and an air-acetylene atomization system (Model No. AA-6800, Shimadzu) to determine the concentrations of K, Ca and P.

## Determination of firmness

Change in firmness or softness are key indicators of fruit ripening. The firmness of mango was evaluated using hand-feel method with predefined numerical grading scale as described by [52]. The numerical grading scale from 1–5 was applied, where 1 = hard (no perceptible deformation, 2 = rubbery (slight deformation), 3 = sprung (flesh deforms by approximately 2–3 mm under extreme thumb pressure), 4 = firm soft (whole fruit deforms under moderate hand pressure), and 5 = soft (whole fruit deforms with slight hand pressure). This method has also been adopted by Holmes et al. [53] and Grice et al. [54]. Although the non-instrumental assessment may introduce some operator variability and limitations in uniformity of measurement, assessment was performed by the same trained evaluator throughout the experiment to maintain consistency.

## Determination of shelf life

The shelf life of both control and treated fruits was determined during. Storage conditions were maintained at a temperature of 28–32°C and a relative humidity of 85–95%.

## Determination of other visual characters

The maturity stage, taste, color and other physiological changes were observed visually.

## Statistical analysis

The collected data were compiled and organized in a proper form for statistical analysis. Results are expressed as a mean ± standard error (SE). Statistical analysis was performed using one-way analysis of variance (ANOVA) in SPSS version 22 (IBM Corp., USA). Mean differences among treatments were adjusted using Duncan's Multiple Range Test (DMRT) at a significance level of $p \leq 0.05$.

# Results and discussion

## Properties of PAW

Table 1 displays the properties of PAW prepared by the experimental setup with air and oxygen plasma with a treatment duration of 10 min. It is seen from the (Table 1) that a significant amount of ROS and RNS are produced in air discharge plasma with respect to that of oxygen.

## Effect of PAW treatments on the incidence (%) of anthracnose

The plasma treatment showed a significant effect ($P \leq 0.05$) on the incidence (%) of anthracnose, as shown in Table 2. Disease incidence was recorded on the 6th, 7th, 8th, 9th, and 10th days of storage at room temperature. In most of the cases, the highest incidences (17%, 18%, 40%, 80%, and 84%) were observed in the control treatment ($T_0$). In contrast, no disease incidence (up to 8th day) or the minimal incidences (18% and 20%) were observed under $T_1$ treatment (10 min air treated PAW) in the mango variety Khirsapat. Treatment $T_3$ (chemical fungicide) exhibited a disease incidence of 41%. However, excessive use of chemical fungicides adversely affects food quality and the environment [24]. Again, 10 min air treated PAW significantly reduced the incidence of anthracnose in the mango variety Fazlee. Up to 8 days of storage, no mangoes were found to be infected with anthracnose. On the 9th and 10th days, only 20% and 22% disease incidences were recorded under the treatment $T_1$, whereas the control showed the highest incidences (60% and 89%), respectively. Overall, the results demonstrated that 10 min air treated PAW was superior to both oxygen treated PAW ($T_2$) and conventional chemical treatment ($T_3$) in controlling post-harvest anthracnose. Plasma treatment effectively inactivated phytopathogenic fungi present on the fruit surface, resulting in a marked reduction in disease incidence. This drastic reduction in anthracnose may be attributed to the bio-chemical properties of reactive oxygen species, mainly $H_2O_2$, and nitrogen species (RONS). Because, the highly reactive $H_2O_2$ along with other reactive species, directly inactivate those fungi that are existed in the fruit surface, be diffused into the fruit coat and be retained there up to their lifetimes, and subsequently prohibit further activation of fungi. Similar results were reported by Zhao et al. [28], who observed the antibacterial

**Table 1. Properties of plasma activated water (PAW) treated with air and oxygen for the duration of 10 min.**

| Treatment condition | Concentration (mg/L) | | | | pH |
|---|---|---|---|---|---|
| | $NO_2^-$ | $NO_3^-$ | $H_2O_2$ | $O_3$ | |
| Air | 41.69 | 12.62 | 22.82 | 0.3 | 3.4 |
| $O_2$ | 0.0 | 0.07 | 1.25 | 0.0 | 7.8 |

**Table 2. Effect of PAW treatments on incidence (%) of anthracnose of mango var. Khirsapat and Fazlee upto 10 days of storage.**

| Treatment | Khirsapat | | | | | Fazlee | | | | |
|---|---|---|---|---|---|---|---|---|---|---|
| | Disease incidence of anthracnose | | | | | Disease incidence of anthracnose | | | | |
| | 6th day | 7th day | 8th day | 9th day | 10th day | 6th day | 7th day | 8th day | 9th day | 10th day |
| $T_0$ | 17±1.15[a] | 18±1.15[a] | 40±1.15[a] | 80±0.58[a] | 84±1.73[a] | 20±2.31[a] | 26±1.15[a] | 28.33±1.67[a] | 60±0.58[a] | 89±0.58[a] |
| $T_1$ | 00±00[b] | 00±00[b] | 00±00[c] | 18±0.58[c] | 20±1.15[c] | 00±00[b] | 00±00[c] | 00±00[c] | 20±0.58[d] | 22±1.15[c] |
| $T_2$ | 00±00[b] | 00±00[b] | 20±0.58[b] | 40±1.15[b] | 40±1.73[b] | 20±0.58[a] | 21±0.58[b] | 23±0.58[b] | 24±1.15[c] | 40±0.88[b] |
| $T_3$ | 00±00[b] | 00±00[b] | 20.66±1.15[b] | 40±1.15[b] | 41±1.15[b] | 20±1.73[a] | 22±1.15[b] | 23.33±1.45[b] | 40±1.15[c] | 40±0.58[b] |
| Level of significance | *** | *** | *** | *** | *** | *** | *** | *** | *** | *** |

In column, means having similar letter(s) or without letter are identical and those dissimilar letters differed significantly. LSD at 0.05 levels of probability, $T_0$ = Control, $T_1$ = Air treated PAW for 10 minutes, $T_2$ = Oxygen treated PAW for 10 minutes and $T_3$ = Diphenoconazole+azoxystrobin@0.6ml/LH$_2$O, *** = Significant at 0.1% level of probability, ± = standard error.

efficiency of RONS as existed in PAW. Likewise, non-thermal atmospheric plasma (NTAP) has been effectively used to decontaminate microorganisms, including fungi and bacterial spores, and is considered cost-effective [25]. Further, Impe et al. [55] reported that cold plasma has the potentiality to inactivate vegetative cells and fungal spores.

## Effect of PAW treatments on severity (%) of anthracnose

A significant difference ($P \leq 0.05$) was found among the plasma treatments in reducing the severity of anthracnose in mango (Table 3). As evident from the table, the control treatment ($T_0$) showed the highest disease severities (3%, 10%, 20%, 43.33%, and 61.76%) across all sampling dates. In contrast, only 2% and 2.33% disease severity were recorded for air treated PAW ($T_1$) on the 9th and 10th days, respectively. Following $T_1$, treatment $T_2$ (oxygen treated PAW) also demonstrated a substantial controlling effect on the severity of anthracnose in Khirsapat. For the Fazlee, fruits treated with $T_1$ showed 100% protection against anthracnose up to the 8th day of storage. Conversely, the highest severity (30%) was recorded in $T_0$. Likewise, on the 9th and 10th days, the lowest disease severities (0.23% and 3.67%) were observed under the treatment $T_1$, whereas $T_0$ showed severities of 38.33% and 56.67%, respectively. The long-lived H$_2$O$_2$, as existed in PAW mentioned earlier, plays critical role in inactivating anthracnose-caused fungal spores and thereby reduce disease severity. These outcomes are consistent with the results of Wu et al. [56], who stated that the long-lived aqueous-phase reactive oxygen and nitrogen species (RONS) play a crucial role in microbial inactivation. Besides, they noted that the inactivation rate of *C. gloeosporioides* was higher in air treated PAW rather than oxygen treated PAW. Wu et al. [57] found

**Table 3. Effect of PAW treatments on severity of anthracnose mango var. Khirsapat and Fazlee upto 10 days of storage.**

| Treatment | Khirsapat | | | | | Fazlee | | | | |
|---|---|---|---|---|---|---|---|---|---|---|
| | Disease severity of anthracnose | | | | | Disease severity of anthracnose | | | | |
| | 6thday | 7th day | 8th day | 9th day | 10thday | 6th day | 7th day | 8th day | 9th day | 10thday |
| $T_0$ | 3.0±1.52[a] | 10±2.88[a] | 20[a]±5.78[a] | 43.33±6.67[a] | 61.67±7.26[a] | 8.33±1.67[a] | 18.33±4.4[a] | 30±5.77[a] | 38.33±7.26[a] | 56.67±8.82[a] |
| $T_1$ | 0.0±0.0[b] | 0.0±00[b] | 0.0±0.0[b] | 2.00±1.50[b] | 2.33±1.33[c] | 0.0±0.0[b] | 0.0±0.0[b] | 0.0±0.0[b] | 0.23±0.13[c] | 3.67±1.33[c] |
| $T_2$ | 0.0±0.0[b] | 0.0±00[b] | 2.0±1.50[b] | 8.33±1.67[b] | 15.00±2.89[bc] | 0.27±0.12[b] | 2.67±1.20[b] | 7.33±1.20[b] | 13.33±1.67[b] | 18.33±1.67[c] |
| $T_3$ | 0.0±0.0[b] | 0.0±00[b] | 3.67±1.33[b] | 11.67±4.41[b] | 21.67±7.26[b] | 0.37±0.13[b] | 3.67±1.33[b] | 8.33±1.67[b] | 20.00±2.89[b] | 36.67±3.33[b] |
| Level of significance | *** | *** | *** | *** | *** | *** | *** | *** | *** | *** |

In column means having similar letter(s) or without letter are identical and those dissimilar letters differed significantly LSD at 0.05 levels of probability, $T_0$ = Control, $T_1$ = Air treated PAW for 10 minutes, $T_2$ = Oxygen treated PAW for 10 minutes and $T_3$ = Diphenoconazole+azoxystrobin@0.6ml/LH$_2$O, *** = Significant at 0.1% level of probability, ± = standard error.

in another investigation that mango treated with dielectric barrier discharged plasma exhibited damage to the subcellular structure of *C. asianum* spore through plasma-derived ROS oxidation, resulting in effective control of anthracnose disease. On the other hand, Phan et al. [58] demonstrated that Nam Dok Mai mangoes treated with gliding arc discharge non-thermal plasma, where OH radicals and $H_2O_2$ were produced, suppressed mycelial growth and consequently delayed anthracnose development. However, it is observed that the air treated PAW provides better inactivation efficiency against anthracnose than that of either oxygen treated PAW or chemical fungicides. Because, air treated PAW contains higher ROS and RNS.

### Effect of PAW treatments on the incidence (%) of stem-end rot

The treatments used in this study showed significant variation ($P \le 0.05$) in the incidence of stem-end rot in both the mango varieties Khirsapat and Fazlee (Table 4). After 10 days of storage, the highest incidence of stem-end rot 60% and 80% were recorded, respectively in Khirsapat and Fazlee in $T_0$. In contrast, the lowest incidence 20% and 0% were recorded in Khirsapat and Fazlee, respectively, when treated with $T_1$. It is interesting to mention that no stem-end rot infection was detected in $T_1$-treated fruits up to 9th day, whereas the control group exhibited the highest level of infection.

The results obtained in this study coincide with the findings of Thirumdas et al. [59], who reported that ROS and RNS present in the PAW possess strong antimicrobial properties, making PAW effective for microbial decontamination. In addition, the presence of $H_2O_2$ in PAW has been shown to effectively inactivate Newcastle virus after 30 min of exposure [60]. According to Xu et al. [61], RONS in PAW interact with the microbial cell walls and membranes, subsequently damaging the cell membrane and leading to microbial inactivation. Collectively, PAW can effectively inactivate fungal pathogens while maintaining overall fruit quality.

### Effect of PAW treatments on the severity (%) of stem-end rot

The severity of stem-end rot in mango was significantly influenced by the application of different plasma treatments at 6th, 7th, 8th, 9th, and 10th days of storage (Table 5). In Khirshapat, the highest severities of stem-end rot (4.33%, 40%, 56.67%, 78.33% and 90%) were observed in $T_0$. No disease symptoms were observed for $T_1$ treatment up to the 9th day of storage, but 15% stem-end rot severity was found at 10th day. In contrast, in Fazlee, the $T_1$ treatment remained completely free from stem-end rot throughout the 10-day storage period. The findings dictate that PAW is capable of reducing the severity of stem-end rot. Further, the findings show a significant difference among the air treated PAW, oxygen treated PAW, and chemical fungicide, where the air treated PAW provides the highest efficiency of 15% in case of Khirsapat, compared to either oxygen treated PAW or chemical fungicide. Besides, no severity (0.00%) of stem-end rot is found in case of Fazlee.

**Table 4. Effect of PAW treatments on incidence (%) of stem-end rot.**

| Treatment | Disease incidence (%) of stem-end rot | | | | | | | | | |
|---|---|---|---|---|---|---|---|---|---|---|
| | Khirsapat | | | | | Fazlee | | | | |
| | 6th day | 7th day | 8th day | 9th day | 10th day | 6th day | 7th day | 8th day | 9th day | 10th day |
| $T_0$ | 20±1.73ᵃ | 21.33±1.33ᵃ | 23±1.15ᵃ | 40.33±0.33ᵃ | 60±1.15ᵃ | 20±0.58ᵃ | 20.67±0.67ᵃ | 25±2.89ᵃ | 60±1.15ᵃ | 80.00±1.73ᵃ |
| $T_1$ | 0.0±0.0ᵇ | 0.0±0.0ᵇ | 0.0±0.0ᵇ | 0.0±0.0ᶜ | 20±1.15ᶜ | 0.0±0.0ᵇ | 0.0±0.0ᵇ | 0.0±0.0ᵇ | 0.0±0.0ᵈ | 0.0±0.0ᵈ |
| $T_2$ | 0.0±00ᵇ | 0.0±0.0ᵇ | 0.0±0.0ᵇ | 20±0.58ᵇ | 21±0.58ᶜ | 0.0±0.0ᵇ | 0.0±0.0ᵇ | 0.0±0.0ᵇ | 20±0.58ᶜ | 20.33±0.33ᶜ |
| $T_3$ | 0.0±0.0ᵇ | 20±1.15ᵃ | 20.33±0.88ᵃ | 20.67±0.67ᵇ | 30±0.58ᵇ | 19.67±0.88ᵃ | 20±1.15ᵃ | 20.67±1.20ᵃ | 30±0.58ᵇ | 40.00±0.58ᵇ |
| Level of significance | *** | *** | *** | *** | *** | *** | *** | *** | *** | *** |

In column, means having similar letter(s) or without letter are identical and those dissimilar letters differed significantly. LSD at 0.05 levels of probability, $T_0$=Control, $T_1$=Air treated PAW for 10 minutes, $T_2$=Oxygen treated PAW for 10 minutes and $T_3$=Diphenoconazole+azoxystrobin @ 0.6 ml/$LH_2O$
***=Significant at 0.1% level of probability, ±= standard error.

**Table 5. Effect of PAW treatments on severity (%) of stem end rot of mango var. Khirsapat and Fazlee.**

| Treatment | Disease severity (%) of stem-end rot | | | | | | | | | |
|---|---|---|---|---|---|---|---|---|---|---|
| | Khirsapat | | | | | Fazlee | | | | |
| | 6th day | 7th day | 8th day | 9th day | 10th day | 6th day | 7th day | 8th day | 9th day | 10th day |
| $T_0$ | 4.33±0.67[a] | 40±5.77[a] | 56.67±4.41[a] | 78.33±4.41[a] | 90.00±5.77[a] | 15.00±2.89[a] | 30.00±5.77[a] | 51.67±6.01[a] | 61.67±7.26[a] | 93.33±3.33[a] |
| $T_1$ | 0.0±0.0[b] | 0.0±0.0[b] | 0.0±0.0[b] | 0.0±00[d] | 15.00±2.88[c] | 0.0±0.0[b] | 0.0±0.0[c] | 0.0±0.0[b] | 0.0±0.0[b] | 0.0±0.0[c] |
| $T_2$ | 0.0±0.0[b] | 0.0±0.0[b] | 0.0±0.0[b] | 20.00±5.77[c] | 20.33±0.33[c] | 0.0±0.0[b] | 0.0±0.0[c] | 0.0±0.0[b] | 4.0±1.0[b] | 8.33±1.67[c] |
| $T_3$ | 0.0±0.0[b] | 5.0±0.0[b] | 10.00±2.89[b] | 40.00±5.77[b] | 70.00±5.77[b] | 5.0±0.0[b] | 16.67±3.33[b] | 35±7.63[a] | 43.33±7.26[a] | 60.00±5.77[b] |
| Level of significance | *** | *** | *** | *** | *** | *** | *** | *** | *** | *** |

In column, means having similar letter(s) or without letter are identical and those dissimilar letters differed significantly. LSD at 0.05 levels of probability, $T_0$=Control, $T_1$=Air treated PAW for 10 minutes, $T_2$=Oxygen treated PAW for 10 minutes and $T_3$=Diphenoconazole+azoxystrobin@0.6ml/L $H_2O$ ***=Significant at 0.1% level of probability, ±= standard error.

Phenomenon regarding disease severity may be attributed as follows: Fazlee has a thinner fruit coat compared to Khirsapat, allowing more reactive species from PAW to diffuse into fruit coat and accumulate, thereby enhancing antifungal activity. Since, one may expect higher concentration of reactive species provide more antifungal activity. Misra et al. [62] demonstrated that strawberries treated with atmospheric cold plasma (ACP) showed a significant reduction in microbial load without any adverse effects on color, texture and flavor of fruit.

## Effect of PAW treatments on the physico-chemical properties of mangos

The use of air treated PAW and $O_2$ treated PAW treatments showed promising effects on the physico-chemical properties of mango (Table 6). The maturity of treated mangoes was significantly influenced by plasma treatments, where the maturity stage refers to the transition from mature-green to fully ripe (Figs 5 and 6). In this study, both mango varieties in the control treatment reached an overripe stage, whereas fruit treated with PAW maintained a desirable ripe stage. The delayed ripening suggests that air treated PAW and oxygen treated PAW may modulate physiological processes associated with fruit maturation. Although ethylene production was not directly measured in this study, previous reports indicate that reactive species in PAW can disrupt ethylene biosynthesis, slow the ripening process and maintain mangoes in a mature state for a longer period [63]. Similarly, Cheng et al. [64] demonstrated that PAW treatments reduced the sensitivity of goji-berry fruits to ethylene, thereby delaying the ripening process and maintaining firmness.

Taste refers the levels of sugars, acids and other flavor compounds present in fruits. The tastes of the two mango verities differed under plasma treatments compared to the control. After ten days of storage, var. Khirsapat, air treated PAW ($T_1$) and oxygen treated PAW ($T_2$) for 10 minutes exhibited a normal sweet taste, whereas the control treatment ($T_0$) was very sweet. Similar results were observed in var. Fazlee. The slower conversion of starches to sugars in PAW-treated fruits likely contributed to the retention of more balanced flavor. It can be assumed that PAW treatment allows gradual sugar accumulation, thereby influencing the overall sweetness of the mango. Mango peel color normally changes from green to yellow during storage. In this study, variation in peel color changes was observed under different plasma treatments in the both mango varieties (Table 6). PAW treated fruits required a longer period to transition from green to yellow compared to control. This delayed color change is likely due to slower degradation of chlorophyll and the interaction of reactive species in PAW with enzymes responsible for color change, thereby preserving the initial green color for longer time. Similar effects of plasma treatments have been reported in other fruits. For example, cold plasma and PAW treatments were shown to preserve quality parameter such as firmness, microbial safety, and color in highly perishable blueberries [65]. In contrast, PAW treatment caused noticeable color changes in fresh-cut radicchio [66]. Several studies, however, reported no significant effect of PAW on color in grapes [67,68], strawberries [36], fresh-cut kiwi [38], pears [39],

**Table 6. Effect of PAW treatments on maturity, taste and color of mango var. Khirsapat and Fazlee up to 10 days of retention.**

| Treatment | Khirsapat | | | Fazlee | | |
|---|---|---|---|---|---|---|
| | Maturity stage | Test | Color | Maturity stage | Taste | Color |
| $T_0$ | Over ripe | Very Sweet | Yellow | Over ripe | Very Sweet | Yellow |
| $T_1$ | Ripe | Sweet | Yellowish green | Ripe | Sweet | Yellowish green |
| $T_2$ | Ripe | Sweet | Yellowish green | Ripe | Sweet | Yellowish green |
| $T_3$ | Over ripe | Very Sweet | Yellow | Over ripe | Sweet | Yellow |

In column, $T_0$ = Control, $T_1$ = Air treated PAW for 10 minutes, $T_2$ = Oxygen treated PAW for 10 minutes and $T_3$ = Diphenoconazole+azoxystrobin@0.6ml/ $LH_2O$.

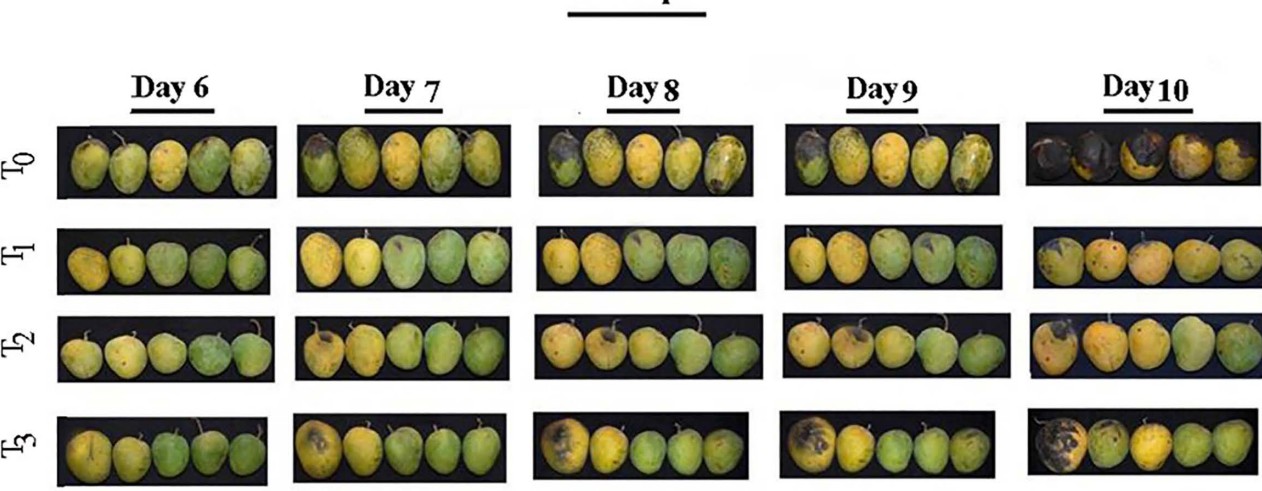

**Fig 5. Physical changes of mango ver. Khirshapat under different plasma activated water treatments at different days.** $T_0$ = Air treated PAW for 10 minutes, $T_2$ = Oxygen treated PAW for 10 minutes, and $T_3$ = Diphenoconazole+azoxystrobin @0.6ml/1L$H_2O$.

fresh-cut apples [40] and shredded salted kimchi cabbage [69]. In summary, while PAW generally does not significantly alter the color of fruits and vegetables, factors such as treatment duration and the specific reactive species generated may critically influence color retention.

## Effect of PAW treatments on the physiological weight loss (%) and moisture content (%)

Physiological weight loss and moisture content of mangoes varied significantly under different plasma treatments (Fig 7A and 7B). In the Khirsapat variety, the highest total weight loss was found in the control (7.69%) treatment, whereas the lowest weight loss (3.12%) was recorded in the air treated PAW for 10 min ($T_1$). Likewise, in the Fazlee variety, the lowest (8.36%) and highest (13.19%) weight loss in mango were recorded in $T_1$ and $T_0$ treatments, respectively. The reactive species present in PAW may suppress enzymatic activities by inactivating surface microorganisms, resulting in reduced physiological weight loss. These findings are consistent with those of Cong et al. [70], who reported that PAW treatment effectively reduced weight loss in goji berries during storage and prolonged their shelf-life. Shi et al. [71] also reported lower weight loss in $O_3$-MNB treated fresh parsley compared with the control. Likewise, Chen et al. [39] observed

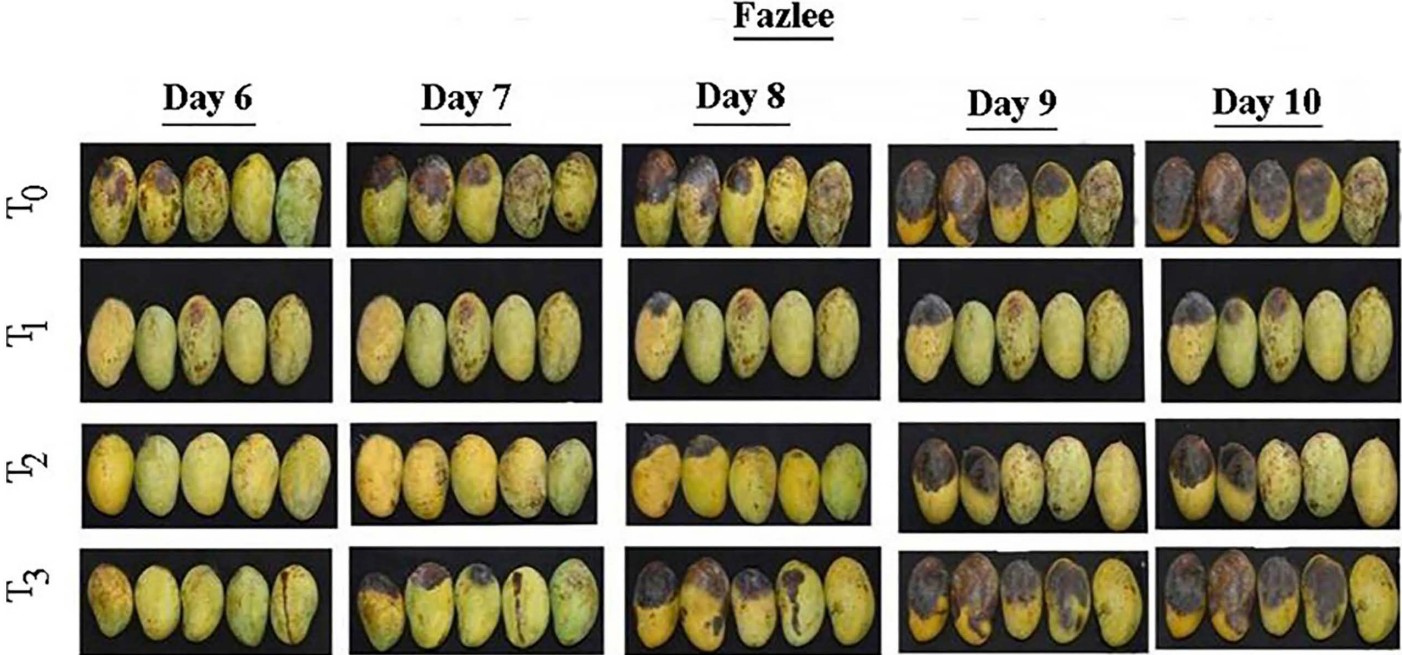

**Fig 6. Physical changes of mango var. Fazlee under different plasma activated water treatments at different days.** $T_0$=Control, $T_1$=Air treated PAW for 10 minutes, $T_2$=Oxygen treated PAW for 10 minutes and $T_3$=Diphenoconazole+azoxystrobin@0.6ml/L$H_2$O.

significantly reduced weight loss in fresh-cut pears treated with activated water and stored at 4°C for 12 days. The above discussions clearly demonstrate that PAW can effectively reduce weight loss in fruits and vegetables.

Moisture content of the two mango varieties ranged from 72–85%. In the Khirsapat variety, the highest and the lowest moisture contents (77.82% and 72.60%) were observed under the air treated PAW for 10 min ($T_1$) and the control ($T_0$), respectively. Similarly, in the Fazlee variety, the highest moisture content (85.34%) was recorded in the air treated PAW for 10 min ($T_1$), while the lowest value (80.23%) was found in the control ($T_0$). The increase in moisture content may be attributed to the effects of PAW treatment. This finding is in agreement with Obajemihi et al. [72], who reported that PAW-treated cherry tomatoes retained greater firmness and exhibited reduced weight loss, indicating improved moisture preservation.

### Effects of PAW treatments on pH and total soluble solids (TSS)

The effects of plasma treatments on pH and total soluble solids (TSS) of juice extracted from mango were evaluated (Fig 8A and 8B). P$^H$ plays an essential role in the inactivation of microorganisms [73] and has a significant influence on the fruit flavour and shelf life. In the Khirsapat variety, the highest (6.63) p$^H$ value was observed in treatment $T_1$ (air-PAW for 10min), while the lowest pH (4.90) was recorded in the control treatment $T_0$. Similarly, in the Fazlee variety, the air treated PAW for 10min resulted in a pH value of 6.51. The increase in p$^H$ maybe attributed to the formation of RNS species during PAW treatment. A similar increase in pH was reported in guava following plasma treatment. During plasma exposure, high- energy electrons strike with the nitrogen molecules, initiating a chain of reactions that lead to the formation of nitrogen oxides, which may contribute to the elevation of pH in treated samples [74]. Yanclo et al. [3] also demonstrated that mango samples treated with low pressure cold plasma showed an increased concentration of ROS, resulting in rise in p$^H$.

Total soluble solids (TSS) are an indicator of fruit quality and directly related to fruit acidity. Generally, as fruit mature and reach the ripening stage, acidity decreases while TSS increases [75,76]. In the Khirsapat variety, the maximum

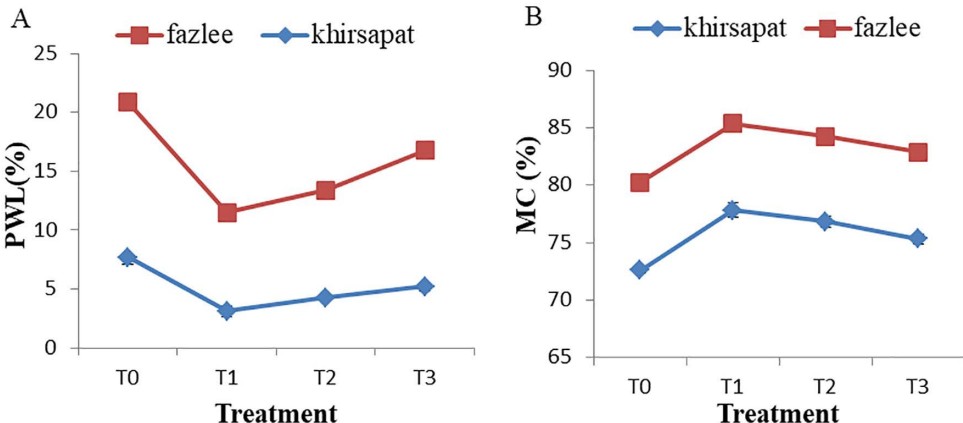

**Fig 7.** **(A) Physiological weight loss (% PWL) and (B) moisture content (% MC) of mango var. Khirsapat and Fazlee due to plasma activated water treatments.** $T_0$ = Control, $T_1$ = Air treated PAW for 10 minutes, $T_2$ = Oxygen treated PAW for 10 minutes and $T_3$ = Diphenoconazole+azoxystrobin @0.6ml/$LH_2O$. Capped knot indicates the standard error difference (SED).

TSS value (18%) was recorded in treatment $T_1$ (air treated PAW for 10 min). A similar trend was observed in treatment $T_3$ (Diphenoconazole+azoxystrobin @0.6 ml/$LH_2O$). In contrast, the lowest TSS value (12.50%) was recorded in the control ($T_0$) treatment. For the variety Fazlee, the highest TSS value (19%) was recorded in the treatment $T_1$ (air treated PAW for 10 minutes), while the second highest value (17.0%) was observed in the treatment $T_2$ compared to the control (14.50%). This increase in TSS may be attributed to the presence of ions, electrons, radicals and UV photons produced during the PAW treatment. These findings are in agreement with those of Ma et al. [35], who found that high level of TSS in Chinese bayberries treated with PAW, possibly due to suppression of the respiratory rate, leading to reduced consumption of sugars and organic acids during storage. Additionally, an increase in TSS was observed in jujube slices treated with cold plasma [77]. Furthermore, cold plasma-generated oxygen radicals (O, OH, $O_3$) can oxidize sugars in fresh-cut mango slices to form carbonyl and carboxyl compounds, thereby contributing to an increase in TSS [78].

## Effects of PAW treatments on firmness and shelf-life

Firmness or softening reflects physiological alternations connected with quality deterioration in fresh fruits [79]. So, firmness is a critical parameter for evaluating fruit quality and shelf-life in mango. The impact of PAW treatment on mango firmness was assessed (Fig 9A and 9B). In the Khirsapat variety, the highest rate of firmness loss (5.76%) was observed in the control treatment ($T_0$), while the lowest rate (4.20%) occurred under the air treated PAW for 10 min ($T_1$). Similarly, in the Fazlee variety, the most rapid firmness degradation (5.79%) was recorded in the control treatment ($T_0$) and the slowest degradation rate (4.06%) was found in the air treated PAW for 10 min treatment ($T_1$). The result suggests that RONS produced by PAW may contribute to firmness retention in ripe mangoes. This observation is consistent with the findings of Liu et al. [80], who stated that PAW treatment preserves cell wall integrity in fruits by regulating the expression of several vital enzymes. Additionally, Cheng et al. [81] demonstrated that PAW-treated blueberries exhibited a significant increase in firmness (36.4%) after 10 days of storage, which was attributed to the controlled solubilization of pectin from water-insoluble to water-soluble forms.

The increase in shelf life observed in PAW-treated mangoes may be associated with reduced physiological weight loss and minimized physical deterioration [82,83]. The highest shelf life was recorded in the $T_1$ treatment (air treated PAW for 10 minutes), with values of 6 days for Khirsapat and 6.33 days for Fazlee. In contrast, the shortest shelf-life (4.0 and 4.67days, respectively) was found in control treatment for both varieties. PAW contains various reactive species (RNS and

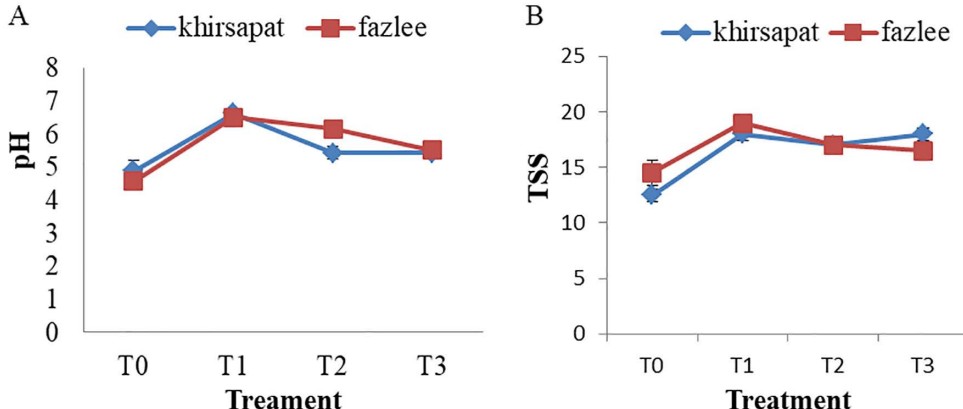

**Fig 8. (A) P$^H$ value and (B) total soluble solids (TSS) of mango var. Khirsapat and Fazlee due to plasma activated water treatments.** $T_0$ = Control, $T_1$ = Air treated PAW for 10 minutes, $T_2$ = Oxygen treated PAW for 10 minutes and $T_3$ = Diphenoconazole+azoxystrobin @0.6ml/LH$_2$O. Capped knot indicates the standard error difference (SED).

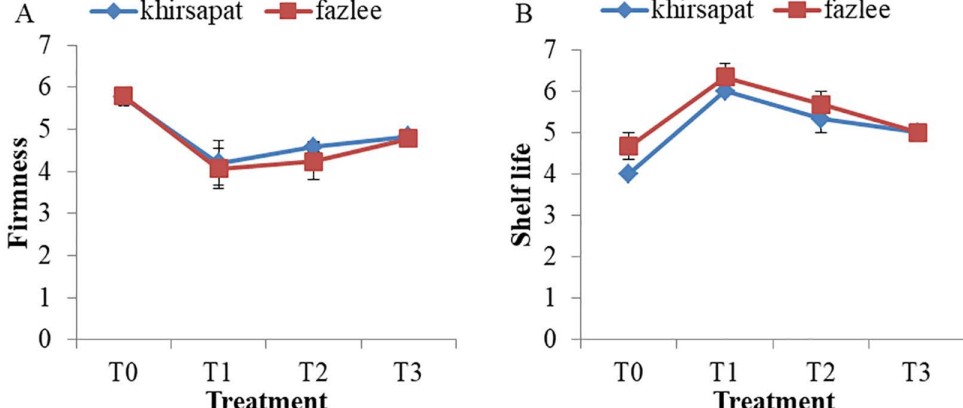

**Fig 9. (A) Firmness and (B) shelf life of mango var. Khirsapat and Fazlee due to plasma activated water treatments.** $T_0$ = Control, $T_1$ = Air treated PAW for 10 minutes, $T_2$ = Oxygen treated PAW for 10 minutes and $T_3$ = Diphenoconazole+azoxystrobin@0.6ml/LH$_2$O. Capped knot indicates the standard error difference (SED).

ROS) that exhibit antimicrobial effects, thereby contributing to prolonged shelf life in ripe mangoes. These findings are in agreement with Yang et al. [84], who reported that PAW treatment delayed quality deterioration and extended the shelf-life of strawberry.

### Effects of PAW treatments on mineral

Significant variation ($p \leq 0.05$) in mineral content was observed in mango pulp following plasma treatments (Fig 10A and 10B). The concentrations of minerals such as potassium (K), magnesium (Mg) calcium (Ca) and phosphorus (P) differed significantly as a result of post-harvest plasma treatments. In the Khirsapat variety, the highest levels of K, Ca, Mg, and P (20.58, 27.64, 10.04, 19.08 mg/100g, respectively) were recorded under the air treated PAW for 10 min ($T_1$), compared with the control ($T_0$), which showed lower values (16.99, 24.97, 7 and 16.76 mg/100g). Similarly, in the Fazlee variety, the $T_1$ treatment significantly increased the contents of K, Ca, Mg and P (18.64, 9.87, 5.66, and 15.78 mg/100g, respectively),

whereas, the lowest mineral concentrations (17.34, 7.76, 3.38 and 13.38 mg/100g) were found in the control treatment ($T_0$). Cold plasma acts as a mild abiotic stress, activating physiological defense mechanisms in fruits. This stress response can stimulate biochemical pathways and alter nutrient metabolism, helping maintain or increase the concentrations of certain nutrients and minerals [85]. PAW, enriched with reactive oxygen and nitrogen species (RONS), may retard enzymatic activity during the ripening process, thereby contributing to the preservation of potassium content [86]. Furthermore, reactive plasma species may disrupt the ethylene biosynthesis, resulting in delayed ripening. This delay may enhance calcium retention by promoting calcium- pectate linkages that stabilize cell wall structure, leading to improve calcium preservation in fruit tissues [87].

## Effects of PAW treatments on the vitamin C content

Vitamin C is regarded as one of the most valued quality attributes of fruit juices. The effect of plasma treatments on the vitamin C content of the two mango varieties (Khirsapat and Fazlee) is presented in Fig 11. The control treatment ($T_0$) showed the highest vitamin C content (20.19 and 28.53 mg/100g), whereas the air treated PAW for 10 minutes ($T_1$) resulted in the lowest value (16.07 and 25.10 mg/100 g) for both mango varieties. Although PAW treatment enhanced several postharvest quality attributes, including microbial reduction, firmness retention, and delayed ripening, a significant decrease in vitamin C content was observed in both mango varieties. Since vitamin C is a key indicator of nutritional quality; therefore, its reduction represents an important trade-off between nutritional preservation and improvements in physicochemical quality. The possible mechanism of the reduction in vitamin C may be attributed to chemical reactions induced by RONS generated during PAW treatment. Similar observations were made by Xu et al. [88] who found that vitamin C concentration in orange juice decreased following treatment with high-voltage atmospheric cold plasma for 120 s. The decrease in vitamin C content can be influenced by the oxidative reactions involving free radicals and ozone generated in the course of treatment. Additionally, treatment duration appears to be a critical factor influencing ascorbic acid degradation [89]. These results suggest that careful optimization of PAW treatment parameters is required to achieve a balance between extending shelf life and maintaining nutritional quality, particularly with respect to vitamin- C.

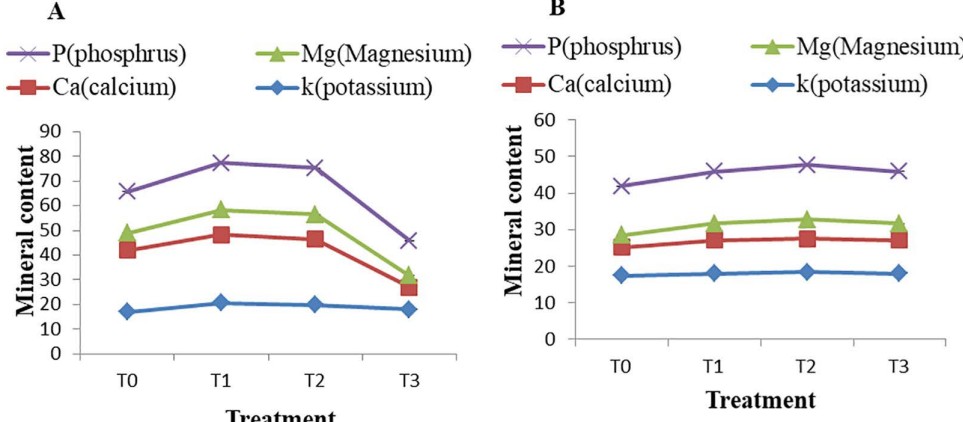

**Fig 10. Mineral content of mango var. (A) Khirsapat and (B) Fazlee under plasma activated water treatment.** $T_0$ = Control, $T_1$ = Air treated PAW for 10 minutes, $T_2$ = Oxygen treated PAW for 10 minutes and $T_3$ = Diphenoconazole+azoxystrobin @0.6ml/L$H_2$O. Capped knot indicates the standard error difference (SED).

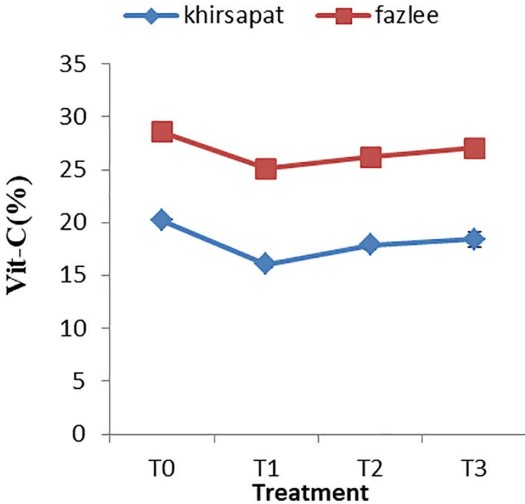

**Fig 11. Vit-C (mg/100g) content of mango var. Khirsapat and Fazlee due to plasma activated water treatments.** $T_0$=Control, $T_1$=Air treated PAW for 10 minutes, $T_2$=Oxygen treated PAW for 10 minutes, $T_3$=Diphenoconazole+azoxystrobin @0.6ml/$LH_2O$. Capped knot the standard error difference (SED).

## Conclusion

The findings of the present study demonstrate that the plasma-activated water (PAW) shows promising post-harvest management of anthracnose and stem-end rot diseases in mango. The results indicate that PAW treatment may reduce disease incidence and severity while contributing to the maintenance of post-harvest quality parameters, including physicochemical properties, bio-active compounds (such as minerals, Vit-C and total soluble solids), firmness, physiological weight loss, and shelf life. In particular, air-treated PAW applied for 10 minutes showed good results under laboratory conditions. This emphasizes the importance of precision application of PAW treatments to strike a delicate balance between positive effects and potential drawbacks. However, these results were obtained under laboratory conditions, and their direct applicability to commercial or industrial-scale operation has not been evaluated. Therefore, the broader practical application of PAW for post-harvest mango disease management cannot yet be confirmed. Additionally, as the reactive oxygen and nitrogen species (RONS) in PAW are short-lived due to limited stability in aqueous system, immediate use after production appears necessary. Further research is required to elucidate the underline mechanisms of PAW action and to assess industrial feasibility, optimization of treatment parameters and long term economic and operational viability before recommending commercial adoption.

## Supporting information

**S1 Appendix. Effect of PAW treatments on incidence (%) of anthracnose of mango var. Khirsapat and Fazlee upto 10 days of storage, replication, mean value, standard error.**
(DOCX)

**S2 Appendix. Effect of PAW treatments on severity of anthracnose mango var. Khirsapat and Fazlee upto 10 days of storage. replication, mean value, standard error.**
(DOCX)

**S3 Appendix. Effect of PAW treatments on incidence (%) of stem-end rot, replication, mean value, standard error.**
(DOCX)

**S4 Appendix. Effect of PAW treatments on severity (%) of stem end rot of mango var. Khirsapat and Fazlee, replication, mean value, standard error.**
(DOCX)

**S5 Appendix. Physiological weight loss (%) and Moisture content (%), replication, mean value, standard error.**
(DOCX)

**S6 Appendix. P$^H$ and total soluble solids (TSS), replication, mean value, standard error.**
(DOCX)

**S7 Appendix. Firmness and shelf-life of mango, replication, mean value, standard error.**
(DOCX)

**S8 Appendix. Mineral content of mango var. Khirsapat, replication, mean value, standard error.**
(DOCX)

**S9 Appendix. Mineral contents of mango var. Fazlee, replication, mean value, standard error.**
(DOCX)

**S10 Appendix. Vitamin-C content, replication, mean value, standard error.**
(DOCX)

## Acknowledgments

We would like to thank the authority of the Plant Pathology Laboratory, Department of Agronomy and Agricultural Extension and Plasma Engineering Laboratory of the Department of Electrical and Electronic Engineering, University of Rajshahi for providing all the necessary laboratory facility for conducting this research. A special thanks goes to Mr. Atik Sheikh for schematic illustration of the mango treatment reactor.

## Author contributions

**Conceptualization:** Mst. Habiba Tamanna, Showdia Sarmin, Umme Afroza Irin, Mst. Khadiza Khatun, Mamunur Rashid Talukder, Md. Mahmodol Hasan.

**Data curation:** Mst. Habiba Tamanna, Mst. Khadiza Khatun, Md. Mahmodol Hasan.

**Formal analysis:** Mst. Habiba Tamanna, Showdia Sarmin, Mst. Khadiza Khatun, Md. Mahmodol Hasan.

**Investigation:** Mst. Habiba Tamanna, Showdia Sarmin, Umme Afroza Irin, Mst. Khadiza Khatun, Md. Mahmodol Hasan.

**Methodology:** Mst. Habiba Tamanna, Umme Afroza Irin, Mamunur Rashid Talukder, Md. Mahmodol Hasan.

**Supervision:** Mamunur Rashid Talukder, Md. Mahmodol Hasan.

**Validation:** Mst. Habiba Tamanna, Umme Afroza Irin, Mst. Khadiza Khatun, Md. Mahmodol Hasan.

**Visualization:** Md. Mahmodol Hasan.

**Writing – original draft:** Mst. Habiba Tamanna, Showdia Sarmin, Umme Afroza Irin, Mamunur Rashid Talukder, Md. Mahmodol Hasan.

**Writing – review & editing:** Mst. Habiba Tamanna, Mamunur Rashid Talukder, Md. Mahmodol Hasan.

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
