## [Decision Letter · Decision Letter 0]

21 Jan 2026

Dear Dr. Hasan,

Thank you for submitting your manuscript to PLOS ONE. After careful consideration, we feel that it has merit but does not fully meet PLOS ONE’s publication criteria as it currently stands. Therefore, we invite you to submit a revised version of the manuscript that addresses the points raised during the review process.

We look forward to receiving your revised manuscript.

Kind regards,

Guadalupe Virginia Nevárez-Moorillón, Ph.D.

Academic Editor

PLOS One

Journal Requirements:

2. We note that your Data Availability Statement is currently as follows: All relevant data are within the manuscript and in Supporting Information files.

3. Please ensure that you refer to Figure 7 and 8 in your text as, if accepted, production will need this reference to link the reader to the figure.

Reviewers' comments:

Reviewer's Responses to Questions

**Comments to the Author**

1. Is the manuscript technically sound, and do the data support the conclusions?

Reviewer #1: Yes

Reviewer #2: Yes

2. Has the statistical analysis been performed appropriately and rigorously?

Reviewer #1: Yes

Reviewer #2: Yes

3. Have the authors made all data underlying the findings in their manuscript fully available?

Reviewer #1: Yes

Reviewer #2: Yes

4. Is the manuscript presented in an intelligible fashion and written in standard English?

Reviewer #1: Yes

Reviewer #2: Yes

Reviewer #1: the paper entitled 'Innovative use of plasma-activated water for sustainable post-harvest disease control

and shelf-life enhancement in mango' is dealing with an important topic which is he shelf-life of mango fruits

The paper is generally fine I have some comments arranged by section to improve the quality and reader quality of the manuscript

1/Abstract: the section is fine summarizes the core findings but lacks further numerical data from the experiment

2/Introduction: The section provides good context on mango importance as well as the main postharvest disease problems linked to this crop

L74; kindly correct the sentence (There are a very few reports have been focused on the effects of plasma)

3/Materials and methods: the section is reproducible and written adequately adequately

the disease incidence was inspected every day according to the authors by the table says at selected days explain why?

4/ Results and discussion: Extensive data is presented, and the discussion attempts to link results to literature and propose mechanisms.

5/Conclusion: well written section

Reviewer #2: Revise the title to specify plasma activated water mechanism, treatment duration, and explicitly indicate mango storage conditions.

The abstract lacks quantitative outcomes for disease reduction, shelf life extension, firmness, and vitamin C changes; include key numerical values.

Introduction is descriptive but unfocused; clearly frame the postharvest disease control problem with a defined technological limitation.

Literature review is outdated and regional; incorporate recent high impact PAW postharvest studies for comparative scientific context.

Research gap is weakly articulated; explicitly state what mechanistic or applied knowledge PAW adds beyond existing plasma studies.

Objectives are broad and repetitive; reformulate into specific, measurable outcomes linked to disease, quality, and storage duration.

No hypothesis is stated; authors must clearly define testable hypotheses regarding PAW effects on pathogens and fruit physiology.

Experimental design lacks justification for sample size and replication; provide power rationale or cite comparable postharvest studies.

Plasma reactor description is excessive yet incomplete; include electrical parameters, discharge power, and PAW physicochemical characterization.

Chemical control selection is unjustified; explain why this fungicide was chosen and discuss regulatory relevance for comparison.

Several quality attributes are visually assessed; subjective evaluations require instrumental validation or must be clearly qualified.

Methods cite AOAC protocols without procedural detail; include sample preparation, replication, and calibration information explicitly.

Firmness measured by hand feeling is unacceptable; replace with instrumental texture analysis or justify methodological limitations.

Statistical analysis lacks detail; report exact p values, assumptions testing, and justification for using DMRT.

Results largely restate tables; strengthen interpretation by linking reductions in disease incidence to plasma induced mechanisms.

Discussion selectively cites supportive literature; critically compare PAW efficacy against chemical treatment and oxygen plasma results.

Claims regarding ethylene suppression are speculative; no ethylene or respiration measurements were conducted to support conclusions.

Vitamin C degradation contradicts quality improvement claims; authors must reconcile this tradeoff explicitly in discussion.

Conclusion overstates applicability; acknowledge laboratory scale limitations and absence of industrial feasibility assessment.

.

Reviewer #1: **Yes:** Dr. Riadh IlahyDr. Riadh IlahyDr. Riadh IlahyDr. Riadh Ilahy

Reviewer #2: No

---

## [Author Response · Author response to Decision Letter 1]

11 Mar 2026

Response to the Editor and Reviewers

We sincerely thank the Editor and Reviewers for their careful evaluation of our manuscript entitled with "Innovative use of plasma-activated water for sustainable post-harvest disease control and shelf-life enhancement in mango". We appreciate their constructive comments, which have significantly improved the clarity and scientific quality of the manuscript. Please see responses to referees below. All comments have been addressed point by point. Changes in the manuscript are highlighted in red color and page and line numbers are provided to help locate changed text and refer to page and line numbers in the original submission (as sent in the editor’s email), "[PONE-D-25-57271].pdf "

Response to Journal Requirements

Requirement 1: PLOS ONE's style and file naming

Response:

We thank the editor for the guidance. The manuscript has been reformatted according to the PLOS ONE style templates for the main body and title_ authors_ affiliations.

Requirement 2: Data Availability Statement and minimal data set

Response:

All relevant data are within the manuscript and in Supporting Information files.

We confirm that our revised submission now includes the complete minimal dataset (supporting information) required to replicate all results reported in the manuscript. “All raw data underlying the findings of this study, including values behind means and figures, are provided as Supporting Information files.” There are no ethical or legal restrictions on data sharing.

Requirement 3: Reference to Figures 7 and 8

Response:

We have now clearly cited Figures 7 and 8 as “Figs 5 and 6”in the Results and Discussion section where physico-chemical properties of mangos are described. (Page 24, Line 408)

Requirement 4: Recommended citations

Response: There is no requirement to cite recommended works.

5. Review Comments to the Author

Response: Point-by-point responses are provided below:

Reviewer#1

Reviewers Questions Answers

1. Abstract: the section is fine summarizes the core findings but lacks further numerical data from the experiment Response 1.0 We are grateful for this for this valuable comment. The Abstract has been revised to include key quantitative results on disease reduction, shelf-life extension, firmness retention, and vitamin C changes for both mango varieties. (Page2-3, Lines 31-44 )

2. Comment 1.1. Introduction: The section provides good context on mango importance as well as the main postharvest disease problems linked to this crop. L74; kindly correct the sentence (There are a very few reports have been focused on the effects of plasma). Response 1.1. The sentence has been corrected to "Moreover, very limited information is available regarding the effects of PAW on the management of anthracnose and stem end rot, and physicochemical quality attributes of mango during storage” (Page 5, Lines 89-91).

3. Comment 1.2. Materials and methods: the section is reproducible and written adequately the disease incidence was inspected every day according to the authors by the table says at selected days explain why? Response 1.2. We appreciate this significant statement. Disease symptoms were visually observed every day; however, quantitative data were recorded at selected days (6th-10th day) because disease symptoms were not detectable before this period. This clarification has now been added to the Materials and Methods section. “Disease incidence was recorded regularly to monitor the development of infection during storage and to accurately determine the onset of visible symptoms.”

(Page 9, Lines 183–185)

4. Comment 1.3. Results and discussion: Extensive data is presented, and the discussion attempts to link results to literature and propose mechanisms.

Response 1.3. We thank the reviewer for this positive assessment. Results and Discussion section has been strengthened by improving interpretation and linking findings more clearly with plasma-induced antimicrobial mechanisms.

5. Comment 1.4. Conclusion: well written section

Response 1.4. We thank the reviewer for this encouraging comment.

Reviewer#2

1. 2.1 . Revise the title to specify plasma activated water mechanism, treatment duration, and explicitly indicate mango storage conditions. The revised title is “Plasma-activated water: Mechanism and treatment duration for postharvest disease control and shelf-life enhancement of mango under ambient storage”

2. 2.2 . The abstract lacks quantitative outcomes for disease reduction, shelf life extension, firmness, and vitamin C changes; include key numerical values. Response 2.1. Key numerical values for disease reduction, shelf-life extension, firmness degradation, TSS, and vitamin C have been added to the Abstract. (Page2-3, Lines 30-43)

3. 2.2 Introduction is descriptive but unfocused; clearly frame the postharvest disease control problem with a defined technological limitation. Response 2.2. The Introduction section has been revised and reorganized to clearly frame the problem of postharvest disease control by emphasizing the technological limitations of current methods. Furthermore, the need for environmentally friendly alternatives has been highlighted, with particular focus on the potential application of plasma-activated water (PAW) as a sustainable postharvest disease management technology.

Comment 2.3. Literature review is outdated and regional; incorporate recent high impact PAW postharvest studies for comparative scientific context.

Response 2.3. Recent high-impact PAW studies (2017–2025) have been added to the Introduction and Discussion sections to provide updated comparative scientific context.

Comment 2.4. Research gap is weakly articulated; explicitly state what mechanistic or applied knowledge PAW adds beyond existing plasma studies.

Response 2.4. The research gap has now been explicitly stated as: “Although the previous investigations have reported disease reduction in fruits in other countries, to the best of our knowledge, no comprehensive studies have assessed using PAW as a post-harvest treatment for mango in Bangladesh, where disease outbreak and post-harvest losses are predominantly high.” (Page 4-5, Lines 84-87)

Comment 2.5. Objectives are broad and repetitive; reformulate into specific, measurable outcomes linked to disease, quality, and storage duration.

Response 2.5. Objectives have been reformulated into specific, measurable outcomes linked into disease, quality, and storage duration.

Comment 2.6. No hypothesis is stated; authors must clearly define testable hypotheses regarding PAW effects on pathogens and fruit physiology.

Response 2.6. Thank you for the comment. Now the hypothesis has been stated as: “We hypothesize that PAW treatment could significantly reduce disease incidence in mango without causing adverse effects during storage”. (Page 5, Lines 91-93).

Comment 2.7. Experimental design lacks justification for sample size and replication; provide power rationale or cite comparable postharvest studies. Response 2.7. We are grateful to the reviewer for this comment. Sample size and replication have now been justified as: “This experiment was laid out using completely randomized design (CRD), in which treatments were randomly assigned to homogeneous experimental units. Under laboratory conditions, the experimental units were homogenous, therefore, blocking was unnecessary. For this reason, the CRD was appropriate for this experiment.” (Page 6, Lines 107-110).

Comment 2.8. Plasma reactor description is excessive yet incomplete; include electrical parameters, discharge power, and PAW physicochemical characterization.

Response 2.8. Now the section has been improved giving information regarding electrical parameters, discharge power, and PAW physicochemical characterization. (Page 7-8 and 14, Lines 140-151 and 270-280).

Comment 2.9. Chemical control selection is unjustified; explain why this fungicide was chosen and discuss regulatory relevance for comparison. Response 2.9. Justification of chemical selection and its regulatory relevance have been added. (Page 8, Lines 151-158).

Comment 2.10. Several quality attributes (taste, color, maturity stage) are visually assessed; subjective evaluations require instrumental validation or must be clearly qualified. Response 2.10. The respective parameters were assessed visually. Although the non-instrumental technique, may introduce some operator variability and limitations in measurement precision, assessments were performed by the same trained evaluator throughout the experiment to maintain consistency.

Comment 2.11. Methods cite AOAC protocols without procedural detail; include sample preparation, replication, and calibration information explicitly. Response 2.11. Detailed procedures for sample preparation, replication, and calibration have now been added for all AOAC-based analysis. Method of mineral content analysis has also been added. (Page 11-12, Lines 207-243).

Comment 2.12. Firmness measured by hand feeling is unacceptable; replace with instrumental texture analysis or justify methodological limitations. The firmness of mangoes was evaluated using the hand-feel method with a predefined numerical grading scale. This approach has also been adopted by other researchers. Although the non-instrumental technique, may introduce some operator variability and limitations in measurement precision, assessments were performed by the same trained evaluator throughout the experiment to maintain consistency. (Page 13, Lines 249-252).

Comment 2.13. Statistical analysis lacks detail; report exact p values, assumptions testing, and justification for using DMRT. Response 2.13. The Statistical analysis section has been expanded to ANOVA assumptions, p-values and justification for using DMRT. (Page 13, Lines 264-267).

Comment 2.14. Results largely restate tables; strengthen interpretation by linking reductions in disease incidence to plasma induced mechanisms. Response 2.14. The Results and Discussion sections have been revised to emphasize interpretation and plasma-induced mechanisms.

Comment 2.15. Discussion selectively cites supportive literature; critically compare PAW efficacy against chemical treatment and oxygen plasma results. Response 2.15. Critical comparisons between PAW, oxygen plasma, and chemical fungicide treatments have been added. Line 287-292

Comment 2.16. Claims regarding ethylene suppression are speculative; no ethylene or respiration measurements were conducted to support conclusions. Response 2.16. Speculative claims have been removed and revised to state clearly that ethylene and respiration were not directly measured in this study. (Page 24, Lines 408-409).

Comment 2.17. Vitamin C degradation contradicts quality improvement claims; authors must reconcile this tradeoff explicitly in discussion. Response 2.17. This trade-off is now explicitly discussed, explaining that PAW improves physicochemical quality while causing modest vitamin C loss and emphasizing optimization of treatment parameters. (Page 33, Lines 582-587 and 594-697).

Comment 2.18. Conclusion overstates applicability; acknowledge laboratory scale limitations and absence of industrial feasibility assessment. Response 2.18. Thank you, reviewer, for the comment. The conclusion is now revised and the limitation of using PAW has been mentioned in the conclusion. (Page 34, Lines 613-621).

We believe that all reviewer and editorial comments have been fully addressed and that the manuscript has been substantially improved. We sincerely thank the Editor and reviewers for their constructive feedback.

---

## [Decision Letter · Decision Letter 1]

5 Apr 2026

Innovative use of plasma-activated water for sustainable post-harvest disease control and shelf-life enhancement in mango

PONE-D-25-57271R1

Dear Dr. Hasan,

We’re pleased to inform you that your manuscript has been judged scientifically suitable for publication and will be formally accepted for publication once it meets all outstanding technical requirements.

Kind regards,

Guadalupe Virginia Nevárez-Moorillón, Ph.D.

Academic Editor

PLOS One

Additional Editor Comments (optional):

Reviewers' comments:

Reviewer's Responses to Questions

**Comments to the Author**

Reviewer #1: All comments have been addressed

2. Is the manuscript technically sound, and do the data support the conclusions?

Reviewer #1: Yes

3. Has the statistical analysis been performed appropriately and rigorously?

Reviewer #1: Yes

4. Have the authors made all data underlying the findings in their manuscript fully available?

Reviewer #1: Yes

5. Is the manuscript presented in an intelligible fashion and written in standard English?

Reviewer #1: Yes

Reviewer #1: The paper has been well revised.

The comments were considered by the authors

I think the paper is now more suitable for publication compared to the original version

.

Reviewer #1: **Yes:** RIADH ILAHYRIADH ILAHYRIADH ILAHYRIADH ILAHY

---

## [Editor Report · Acceptance letter]

PONE-D-25-57271R1

PLOS One

Dear Dr. Hasan,

I'm pleased to inform you that your manuscript has been deemed suitable for publication in PLOS One. Congratulations! Your manuscript is now being handed over to our production team.

Kind regards,

on behalf of

Dr. Guadalupe Virginia Nevárez-Moorillón

Academic Editor

PLOS One